# Social Humanoid Robots for Children with Autism Spectrum Disorders: A Review of Modalities, Indications, and Pitfalls

**DOI:** 10.3390/children9070953

**Published:** 2022-06-25

**Authors:** Alfio Puglisi, Tindara Caprì, Loris Pignolo, Stefania Gismondo, Paola Chilà, Roberta Minutoli, Flavia Marino, Chiara Failla, Antonino Andrea Arnao, Gennaro Tartarisco, Antonio Cerasa, Giovanni Pioggia

**Affiliations:** 1Institute for Biomedical Research and Innovation (IRIB), National Research Council of Italy (CNR), 98164 Messina, Italy; alfio.puglisi@irib.cnr.it (A.P.); tindara.capri@unime.it (T.C.); stefania.gismondo@irib.cnr.it (S.G.); paola.chila@irib.cnr.it (P.C.); roberta.minutoli@irib.cnr.it (R.M.); flavia.marino@irib.cnr.it (F.M.); chiara.failla@irib.cnr.it (C.F.); antoninoandrea.arnao@cnr.it (A.A.A.); gennaro.tartarisco@irib.cnr.it (G.T.); giovanni.pioggia@irib.cnr.it (G.P.); 2Department of Life and Health Sciences, Link Campus University, Via del Casale di S. Pio V, 44, 00165 Rome, Italy; 3S’Anna Institute, 88900 Crotone, Italy; l.pignolo@isakr.it; 4Pharmacotechnology Documentation and Transfer Unit, Preclinical and Translational Pharmacology, Department of Pharmacy, Health Science and Nutrition, University of Calabria, 87036 Arcavacata, Italy

**Keywords:** autism, robot-assisted therapy, humanoid

## Abstract

Robot-assisted therapy (RAT) is a promising area of translational neuroscience for children with autism spectrum disorders (ASDs). It has been widely demonstrated that this kind of advanced technological tool provides a reliable and efficient intervention for promoting social skills and communication in children with ASD. This type of treatment consists of a human-assisted social robot acting as an intervention mediator to increase competence and skills in children with ASD. Several social robots have been validated in the literature; however, an explicit technical comparison among devices has never been performed. For this reason, in this article, we provide an overview of the main commercial humanoid robots employed for ASD children with an emphasis on indications for use, pitfalls to be avoided, and recent advances. We conclude that, in the near future, a new generation of devices with high levels of mobility, availability, safety, and acceptability should be designed for improving the complex triadic interaction among teachers, children, and robots.

## 1. Introduction

Behavioral treatments are the major tool for reducing comorbidity and disability in children with autistic spectrum disorder (ASD) [1]. Generally, these are focused on maximizing the ability in social communication and skills [2]. Various behavioral approaches have been validated for ASD patients and are classified into: (i) comprehensive applied behavioral analysis-based intensive intervention; (ii) targeted skill-based intervention (training in joint attention, teaching social skills, and social skill training); and (iii) targeted behavioral intervention for anxiety and aggression (cognitive behavioral therapy) [3,4]. These treatments promote the development of several emotional and cognitive skills in ASD children. 

However, individuals with ASD show notable heterogeneity at genetic, behavioral, and neurophysiological levels, which could interfere with the efficacy of these interventions [5]. Moreover, it is well-known that ASD individuals engage more successfully in social interactions if social information is presented in an “attractive” manner [6].

The last two decades have seen the emergence of technology-based therapies, such as robot-assisted therapy (RAT), for improving the treatment of individuals with ASD [7]. Robot-mediated intervention studies have shown positive outcomes in improving (a) joint attention, (b) social communication, (c) imitation, and (d) social behaviors [8]. Interacting with robots as an emulated peer is naturally more attractive because this is based on the observation that the well-known limited eye contact of children with a therapist could be successfully stimulated by a social robot [9].

The RAT approach has two fundamental advantages: (a) the opportunity to record objective data during therapy and (b) the ability of the robot to adaptively “learn” both interindividual differences at one time point and intraindividual differences over time, thus partially overcoming the limitations due to clinical heterogeneity. The former characteristic is important to characterize the behavioral improvement, providing quantitative data about the developmental process [9].

In the last few years, several successful interventions have been developed using the RAT approach [8,9,10,11,12], although a rigorous comparison among technical devices has never been performed. In this review, we provide an explicit description of strengths and limitations for devices employed in clinical trials and generally considered by teachers and therapists as the best tool for their practice: the humanoid robots.

## 2. Social Humanoid Commercial Robots

In this qualitative analysis, we consider only humanoid social robots employed in social skills training for ASD children which are commercially available and already validated in clinical trials. Considering these criteria, we reviewed the characteristics of: (1) NAO (Aldebaran Softbank Robotics, Tokyo, Japan), (2) QTrobot (LuxAI S.A.; Luxembourg); (3) KASPAR (Adaptive Systems Research Group at the University of Hertfordshire, Hatfield, UK); (4) FACE (Enrico Piaggio Center for Robotics and Bioengineering of the University of Pisa, Pisa, Italy); and (5) ZENO (Hanson Robotics, Hong Kong, China).

### 2.1. NAO Robot

NAO (dimensions: 574 × 311 × 275 mm; weight: 5.48 kg) is characterized by a body in plastic with 25 DoF (four joints for each arm; two for each hand; five for each leg; two for the head; and one to control the hips). The internal processor is an Intel Atom E3845, using Linux as an operating system (compatible with Windows and MacOS). NAO can also speak and assure a certain degree of non-verbal communication, capturing a lot of information about the environment using sensors and microphones. In detail, the NAO robot is equipped with: Sonar to interpret the distance to objects or subjects.Tactile sensors on the hands and head.A camera (two OV5640 2592 × 1944) and microphones for voice and facial recognition.Speakers to listen to sounds that can be reproduced by the robot itself.Stepper-motors to represent the robot’s movements.Stepper-motors (see Figure 1) that allow movements very similar to a human being’s prehensile hands.An ethernet and wireless network card.

The user-friendly software embedded in the robot works on Mac, Windows, and Linux platforms, although it is not supported by the latest versions of the MacOsX system. In any case, programming through Choreographer’s proprietary software is very limited, but the C++ and Python APIs are available, allowing the robot to be implemented in mobile or desktop applications.

#### 2.1.1. NAO: Clinical Validation

NAO is the most famous and employed device for promoting emotional and cognitive rehabilitation in children with ASD [11]. Several studies demonstrated its effectiveness as a mediator of behavioral interventions. For instance, Marino et al. [12] conducted the first randomized controlled trial using NAO in a socio-emotional understanding protocol for children with ASD. Fourteen children were randomly assigned to 10 sessions of cognitive behavioral therapy intervention applied in a group setting, either with or without the assistance of NAO. The results demonstrated that children performing with the RAT significantly improved their socio-emotional skills with respect to the control group. Van den Berk-Smeekens et al. [13] conducted a randomized controlled trial using Pivotal Response Treatment (PRT) with and without NAO robot assistance for improving the social skills of children with ASD. Seventy-three children were randomly assigned into three groups (PRT: *n* = 25; PRT + robot: *n* = 25; standard intervention: *n* = 23). The results indicated that the PRT + robot group showed a larger improvement in social communication than the other two groups. 

#### 2.1.2. NAO: Advantages vs. Disadvantages

The main strengths of this device are: (a) autonomy, (b) motion, and (c) clinical validation [14] (Figure 2). With reference to robot autonomy, NAO was used in three operating modes: full-autonomy, semi-autonomy, and Wizard of Oz [15]. In the first mode, the robot autonomously detects a child’s behavior or its eye gazes through tracking devices. In the second mode, the actions of the robot are activated both autonomously or by a therapist or researcher. In the Wizard mode, the researcher or therapist remotely controls the robot’s behavior without the child noticing it. This mode is the most-used both with NAO and other robots. With reference to motion, NAO can provide a large variety of human–robot interactions, increasing the types of actions that the robot and child can make together [14]. Moreover, NAO can walk with high degrees of freedom (DOF). For this reason, NAO seems more human-like than other robots that can move their arms only up and down in a single plane of motion [14]. In regard to clinical validation, as explained in the above section, NAO has been used in several clinical studies and validated for behavioral rehabilitation in children with ASD.

Otherwise, NAO is also characterized by some limitations, such as physical appearance and technological features. NAO’s eyes have colored LED to help children in focusing attention on particular social cues that are necessary for the skill being trained. However, this could represent overstimulation, and it is well-known that sensory overstimulation is a serious problem for many children with ASD [16]. Moreover, NAO cannot express facial emotions, and this may not be helpful for children with difficulties in recognizing human facial emotional expressions [1,2,17]. For this reason, a robot cannot appear both extremely human-like and socially simple [14]. Thus, an alternative option for designers is to create evocative but visually simple robots by implementing an additional screen on the robot’s head to display simple emotional facial expressions (see Section 2.2) Finally, although NAO is equipped with guidelines for safety, it is not possible to anticipate and predict all potential situations that could occur when children and robots interact. Indeed, NAO could create physical damage to children’s hands and fingers, given that it has strong prehensile skills [18,19,20,21].

NAO has been built for improving behavioral intervention in ASD children; however, this robot has also been applied in other clinical domains, such as attention deficit hyperactivity disorder (ADHD), language disorders, and Down’s Syndrome [19,20,21]. Taken together, these studies suggest that NAO has the potential to be translated for the treatment and education of children with different disabilities. 

### 2.2. QTrobot

QTrobot is an expressive little humanoid robot (dimensions: 574 × 311 × 275 mm; weight: 5 Kg), designed and built to assist therapists in teaching new skills (cognitive, social, communication, and emotional) to children with autism or special educational needs (Figure 3). This robot is characterized by high motricity in the neck and hands (DOF: 12). This is equipped with: (a) a face display that can show movies, thus emulating basic emotional expressions; (b) a 3D intel RealSense camera that enables vision and gesture recognition in space, as well as excellent resolution for facial recognition; and (c) microphones to recognize where the sound is coming from and speakers which allow the robot to produce verbal communication or play sounds.

An internal Raspberry PI (QTPI) board controls the motors, displays, and sensors, all connected to a Linux PC (QTPC), which uses ROS to send commands to the Raspberry board. The QTPC and Raspberry board that make up the robot are connected to each other via an internal LAN, allowing easy configuration and programming which can be directly sent (via Web) to the company manufacturer for information exchange. This tool provides an opportunity to translate robot-assisted therapy on the Internet of Things (IoT) data domain.

Programming can be performed using the web app interface provided by the manufacturer, which offers an intuitive block-type utility. This allows routines to be created and executed on the robot using the Android tablets that come with the robot.

For customizing specific behaviors, QTrobot allows the use of the RealSense software. This software has been installed into the robot and allows it to recognize gestures or faces through the assignment of key point data in space. It is possible to write specific commands using Python and C++ that invoke the APIs already installed in the robot’s QTPC. The manufacturer’s site (LuxAI) provides extensive tutorials on hardware and software characteristics.

In detail, the QTrobot robot is equipped with (Figure 3): An 8th Gen quad-core Intel^®^ CoreTM i5/i7 processor up to 4 × 4.5 GHz, up to 32 GB DDR4 RAM, and up to 512 GB M.2 SSD.A camera (RealSenseTM depth camera D435; field of view ≈ 87° × 58° × 95°) and microphones (four digital microphones; supports far-field voice capture; microphones: ST MP34DT01TR-M; sensitivity: −26 dBFS) for voice and facial recognition.Speakers to listen to sounds that can be reproduced by the robot itself (audio amplifier: stereo 2.8 W Class D; speaker frequency rate: 800~7000 Hz).Facial Display (8 inch TFT 800 × 480 LCD).An ethernet and wireless network card.

#### 2.2.1. QTrobot: Clinical Validation

QTrobot is a recently developed social robot. Until now, there are only two studies demonstrating its effectiveness as a mediator of behavioral interventions on children with ASD. Costa et al. [22] have examined the use of QTrobot in long emotional-ability training for ASD, providing restricted evidence of the positive effects of the robot-mediated intervention. In another study, Costa et al. [23] have evaluated the usefulness of QTrobot by assessing children’s attention, imitation, and presence of repetitive and stereotyped behaviors. They obtained significant positive results in all considered parameters. 

#### 2.2.2. Qtrobot: Advantages vs. Disadvantages

The most significant advantages of this device are the physical appearance and some technological features. With reference to physical appearance, QTRobot has more closely related human features, with different levels of motion which allow for an easier identification of social actions and expressions, facilitating the transfer of skills learned in the human–robot context to a human–human interaction [4,24,25,26]. QTrobot is built precisely to a child’s physical dimensions; it moves its arms with multiple DOF. Its display allows the presentation of animated faces and emotional facial expressions combined with arm movements and voice. Concerning technological features, the architecture of QTrobot is characterized by simple programming using Internal software, easy to customize with different behaviors (RealSense) useful for robot-assisted applications in the ASD domain [15]. Furthermore, QTrobot has been developed to be employed in both homes and therapy settings.

The most significant disadvantages are that it has few sensory features and effective usage only with digital tablets (Figure 4). Generally, robots employed in RAT should be able to detect the child’s position in order to orient the child in performing specific actions and responses [27]. QTrobot is only equipped with RealSense which does not allow this kind of interactive spatial evaluation. Moreover, the child–robot interaction is mediated by the use of a digital tablet that could create an overstimulation for the child. Another pitfall is the lack of applications in clinical trials. Nowadays, only two studies evaluated the effectiveness of QTrobot in reducing repetitive and stereotyped behaviors and in increasing joint attention and emotional skills in children with ASD [22,23].

### 2.3. KASPAR

Kaspar is a humanoid social robot (dimensions: 55 × 50 × 45 cm; weight: 15 kg; six DoF on the neck and head, six on the arms, and two in the eyes). Its face is a silicon-rubber mask that can show a range of simplified expressions. This can respond to the touch of children and can move its head, arms, and eyes. This is equipped with tactile sensors (Figure 5), which allow the robot to react as previously defined by software programming.

The programming of the robot is performed through an easy programming interface, but it is very limiting as it does not allow the development of interaction with other devices and platforms.

In detail, the KASPAR robot is equipped with: SENSORS Cameras in eyes. Force-sensing resistor or capacitive touch sensors.ACTUATORS Dynamixel AX-12A robot servos and RC servos.POWER One 12-V 7-Ah lead acid battery, 4 hours of operation.COMPUTING Controlled by external PC via USB. Or wirelessly using on-board mini PC.SOFTWARE Custom Java software. YARP, C++, and Python interfaces optional.DEGREES OF FREEDOM (DOF) 17 (Arm: 4 DoF x 2; Neck: 3 DoF; Mouth: 2 DoF; Eyes: 2 DoF; Eyelids: 1 DoF; Torso: 1 DoF)MATERIALS Fiberglass body; aluminum frame and head parts; silicone rubber face.

#### 2.3.1. KASPAR: Clinical Validation

The KASPAR robot has been employed in several clinical trials to demonstrate its effectiveness as mediator of behavioral interventions on children with ASD. Marinoiu, Zanfir, Olaru, and Sminchisesc [28] have used KASPAR in order to involve 13 children with ASD in different games for helping them to see the world from the robot’s perspective (i.e., the theory of mind). The results have indicated that the robot-assisted therapy using KASPAR can be an effective intervention to improve the theory of mind and visual perspective-taking in autism. Recently, the results of other studies have demonstrated that robot-mediated interventions using the KASPAR robot improved communication, psychomotor functions, social skills, and imitation in children with ASD [29]. Reviews on the effectiveness of KASPAR have highlighted the potential of this robot in interventions for children with ASD [30].

#### 2.3.2. KASPAR: Advantages vs. Disadvantages

The most significant advantages of this device are: (a) less complexity of human-related facial emotion expressions; (b) tactile sensors, and (c) it is easy to customize to autism needs (Figure 6). Kaspar has a realistic face with a less complex actuation system [29], i.e., KASPAR can open and close its mouth, can smile and frown, can move its eyes up/down and left/right, and finally, it can open/close the eyelids. This system reduces the complexity of social stimulus; consequently, KASPAR can be more predictable, less distracting, trustable, and less ambiguous than a human person would be [30]. Differently from other robots, KASPAR is equipped with tactile sensors; this means that children can observe the effect of pressing buttons on Kaspar’s motion, so they can benefit from a turn-taking interaction, given that children with ASD usually tend to not engage in such behavior [16].

The main disadvantage of this device is the limited behavioral reactions. KASPAR cannot walk, grasp, or fetch objects, or make fine gestures with its hands or fingers. Mobility is an important factor that must be controlled during a human–robot interaction, because a good movement capability increases the types of actions that the robot and child can engage in together [14]. Additionally, KASPAR is used in a semi-autonomous way; this means that a few predefined actions can be programmed on the remote control [29,30,31]. This limited autonomy influences the application in rehabilitation settings, as well as the development of scenarios for the child–robot interaction.

### 2.4. FACE (Facial Automaton for Conveying Emotions)

FACE is a passive body with an active head. Thirty servomotors simulate and modulate six basic emotions (anger, happiness, surprise, sadness, disgust, and fear). FACE cannot speak, but through its microphones and cameras, it can analyze the emotional reactions of individuals, react to them, and store all data.

The programming of the robot is performed through scratch programming, which is very simple, even for beginners, but very limiting as it does not allow the development of interaction with other devices and platforms.

In detail, the FACE robot is equipped with: SENSORS External cameras and microphones positioned next to the android (used for teleoperation).ACTUATORS Pneumatic actuators in the face (eyes, forehead, eyebrows, eyelids, and cheeks) and body (neck and shoulder).POWER Standard 110-V/220-V power supplyCOMPUTING Custom server and control infrastructureSOFTWARE Windows OS and Java-based applicationDEGREES OF FREEDOM (DOF) 12MATERIALS Metal skeleton, silicone skin for hands and face, wig made of human and artificial hair.

#### 2.4.1. FACE: Clinical Validation

The FACE robot has been employed in some clinical trials to demonstrate its effectiveness as a mediator of behavioral interventions on children with ASD. A study [32] demonstrated that this device aided in improving imitative skills and shared attention, although a small group of ASD children was enrolled. Another study [33] confirms this preliminary evidence, highlighting that all participants have shown an improvement in their imitation abilities and social communication skills after RAT with FACE. Based on these preliminary data, researchers have suggested that treatment with FACE can develop pragmatic emotional responsiveness in children with ASD.

#### 2.4.2. FACE: Advantages vs. Disadvantages

The main significant advantage of this device is the ability to express realistic emotions (Figure 7). Indeed, the FACE robot has been developed based on biological principles to be a realistic facial display system. The FACE robot has servomotors to control facial movements and a biomimetic proprioceptive system. The motors allow us to express six basic emotions based on feedback from the sensing layer [32,33].

Otherwise, FACE is also characterized by limitations, such as missing motion and mobility. It is unable to express complex emotions combining facial emotional expressions with gestures. Moreover, the lack of mobility and motion reduces the variety of human–robot interactions [14]. Finally, another major disadvantage of this device could be the Uncanny Valley effect [34]. Following Masahiro Mori’s statements [34], it describes the relationship between the human-like appearance of a robot and the emotional response evoked in people. Mori observed that people found robots more appealing the more human they appeared, and this feeling induces positive emotion and reaction. However, this sense of familiarity only worked up to a certain point. When the appearance of humanoid robots moves from a “somewhat human” to “fully human” entity, this provokes uncanny or strangely familiar feelings of revulsion in observers. For this reason, the FACE robot could fall into the uncanny valley in ASD children.

### 2.5. ZENO

Zeno is a humanoid child-size robot with a simple expressive face (dimensions: (0.635 m max height; 6.5 Kg weight). The robot’s face has 8 DOF, 3 DOF for the neck, and 25 DOF for the body where motors are used for simulating facial expressions. The body is equipped with servomotors for the legs, hips and shoulders, and waist. The programming of the robot is performed through an easy programming interface, but it is very limiting as it does not allow the development of interaction with other devices and platforms.

In detail, the ZENO robot is equipped with: SENSORSvTwo 720p, 30fps HD cameras (one in each motorized eye).Three-axis gyroscope, three-axis accelerometer, compass.Twenty one joint load sensors, 30 joint position sensors, two cliff sensors, two ground contact sensors, two infrared obstacle-detection sensors, two bump sensors (feet), grip-load sensors in the hands. Three microphones.ACTUATORS Three Cirrus CS-101 STD 4-gram micro servos. Five Hitec HS-65MG motors (Frubber actuators). Dynamixel RX-64 (legs, hips, shoulders). Dynamixel RX-28 servos (waist).POWER Two 18.5-V lithium-ion batteries, 1 hour of operationCOMPUTING 1 GHz Vortex86DX CPU, 1 GB RAM, Wi-Fi, EthernetSOFTWARE Linux UbuntuDEGREES OF FREEDOM (DOF) 36 (Arms: 12 DoF; Legs: 12 DoF; Waist: 1 DoF; Neck: 3 DoF; Face: 8 DoF)MATERIALS Frubber, plastic, and aluminum

#### 2.5.1. ZENO: Clinical Validation

The ZENO robot has been employed in some clinical trials to demonstrate its effectiveness as a mediator of behavioral interventions on children with ASD. In a study [35], researchers sought to stimulate facial emotion recognition skills in children with ASD, compared to typically developing children (TD). Results indicated no significant difference among groups, although ZENO is able to successfully express six basic emotions. Recently, Lecciso et al. [35] enrolled 12 children with ASD, randomly subdivided into two groups: a robot-based intervention with ZENO and a computer based-intervention. Both types of intervention aimed at improving facial emotion recognition. Results have shown no significant differences between the two groups. Both robot and computer intervention produce similar improvements. Overall, future studies are necessary to validate the use of ZENO in the treatment of ASD. 

#### 2.5.2. ZENO: Advantages vs. Disadvantages

The most significant advantages of this device are facial emotional expression and mobility (Figure 8). ZENO is a child-sized and -shaped robot but with limited expressive abilities (only six basic emotions). However, this capability combined with motion (it can move its arms and legs) gives it a human-like physical appearance. It is known that physical appearance and mobility are two important factors that mediate human–robot interactions [24]. This is essential in the context of ASD, given that one of the major impairments in ASD is emotional understanding and recognition. 

The main disadvantage of this device is the low number of DOF. The motor system of ZENO is limited in its bodily capabilities due to the low number of DOF. This drastically decreases the changes to design human-like actions. 

## 3. Discussion

The establishment of an adequate social robot tool is one of the most important clinical targets aimed at increasing the efficiency of RAT approaches for ASD children. Taken together, the results of the present analysis indicate that the most important factors for human–robot interaction, in the context of treatment for ASD, are physical appearance and mobility. The NAO robot has good mobility, even if it can be dangerous for a child’s fingers, but it is limited in its physical appearance. The QT robot has a social, expressive, and simple appearance with a display for showing facial emotion expression, but it is fixed on a stand, and it cannot walk or roll around in their environment. Other robots included in this narrative evaluation show ambiguous physical features and limited mobility.

Overall, it is extremely difficult to design a robot that is able to conflate a human-like appearance with socially interactive capabilities and imitations of the children’s movements in real time. Several key challenges must be addressed. Within the scenario of social assistive robotics for ASD, the main aims for a child–robot interaction are to elicit joint attention, to encourage imitative behaviors, to promote socio-emotional understanding and facial emotion recognition, and for turn-taking between the child with ASD and the robot. Consequently, the challenge is to design a child-size expressive humanoid robot with good mobility and verbal skills. Thus, the robot should be able to walk, move its arms and legs around the environment, and it should also be safe and socially attractive with a human-like appearance.

From a technological point of view, the perception system of a robot must be able to detect the child’s position and movements, because the child is free to move around the room. Both the NAO and QT robots have a good perception system, whereas other robots are limited in this function. Moreover, the robot must be able to express several and complex emotions, not only basic emotions. These are important factors to promote a greater variety of potential actions between the child and robot and to make the therapeutic session more closely life-like. In this case, only QTrobot can express both simple and complex emotions.

From a researcher’s point of view, robot systems must have numerous capabilities, such as: sensing and interpreting the child’s actions; full autonomy within the experimental scenario setting; collecting and processing data over time; evaluating the interaction in terms of the quantity and quality of behaviors; altering behavior based on parameters chosen by the researcher or experimenter; and flexibility in the programming [25]. Again, the NAO and QT robots are equipped with a platform for researchers; however, further developments are needed in order to make these platforms more flexible.

Summarizing, the key idea is to connect the needs of robot developers, care professionals, researchers, and children to increase the efficiency of a robot-assisted mediated cognitive therapy approach for ASD and to design and develop a robot with high levels of utility, availability, safety, and acceptability. 

## 4. Conclusions

To the best of our knowledge, this is the first rigorous comparison among technical devices showing indications for use, pitfalls to be avoided, and recent advances of the most famous humanoid robots used as an intervention mediator to increase the emotional/cognitive competence and skills in children with ASD. There are several reviews on the RAT approach in ASD, but none focus on the technical features of robots. In accordance with previous studies [15,36], the present analysis suggests that to design and develop meaningful robot-mediated interventions, the robot must address the needs of children with ASD, care professionals, and developers.

The current state-of-the-art for social assistive therapy has not reached its full potential yet in terms of physical appearance and technological features which are the two key aspects evidenced in this review. The most-used robots are employed in a wizard way, increasing the burden of care professionals. Some robots are limited in mobility functions, and they are visually and kinetically simple designs. 

The challenge for the future is to design a new era of child-size expressive humanoid robots to improve the complex triadic interaction among teachers and children with the robots, also considering the entry of Artificial Intelligence algorithms that should induce flexibility and learning capabilities in previously rigid applications.

## Figures and Tables

**Figure 1 children-09-00953-f001:**
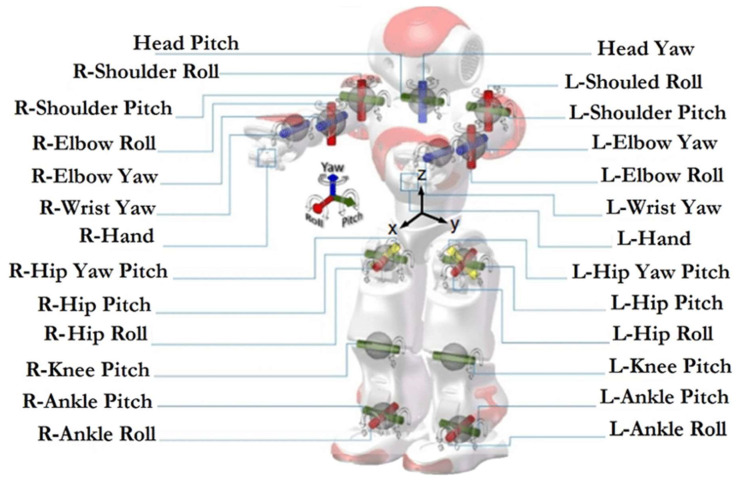
Localization of stepper-motors inside NAO robot.

**Figure 2 children-09-00953-f002:**
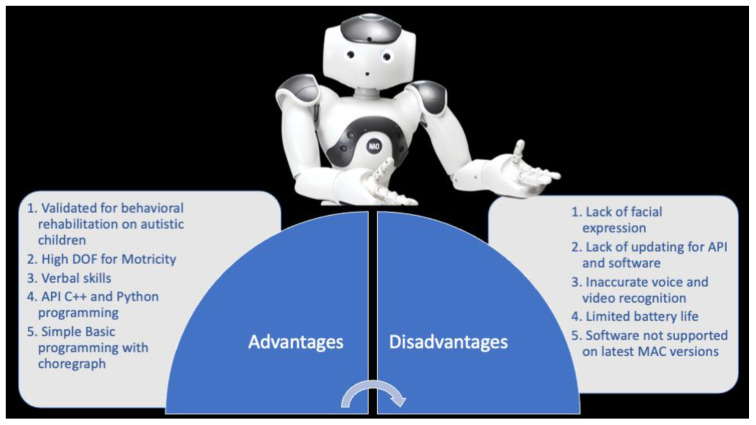
NAO robot.

**Figure 3 children-09-00953-f003:**
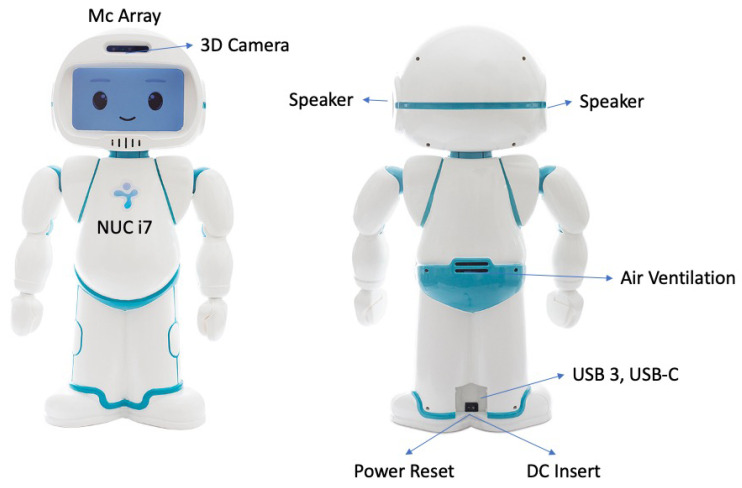
General characteristics of QTrobot.

**Figure 4 children-09-00953-f004:**
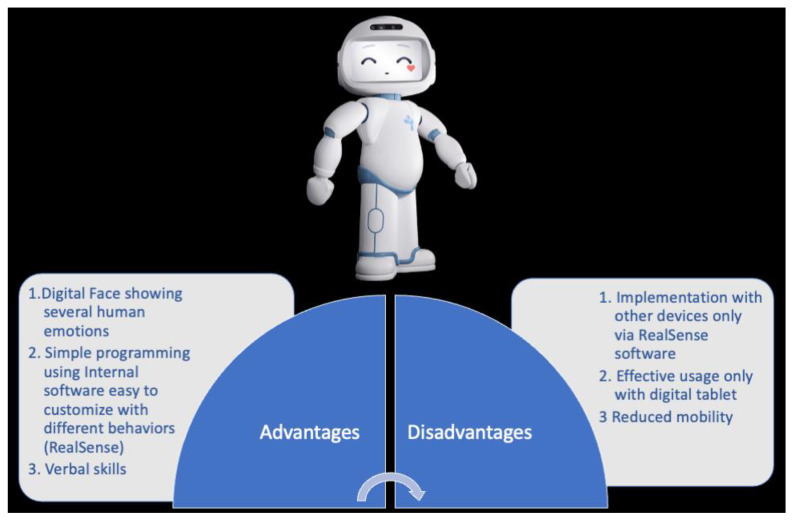
QTrobot.

**Figure 5 children-09-00953-f005:**
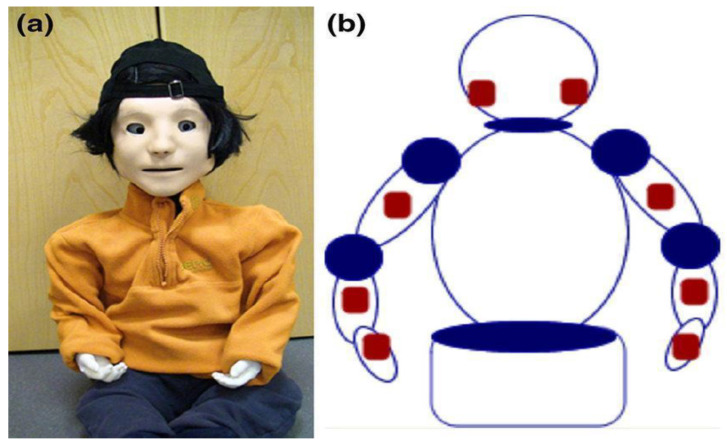
KASPAR robot (**a**) and sensors localization (**b**).

**Figure 6 children-09-00953-f006:**
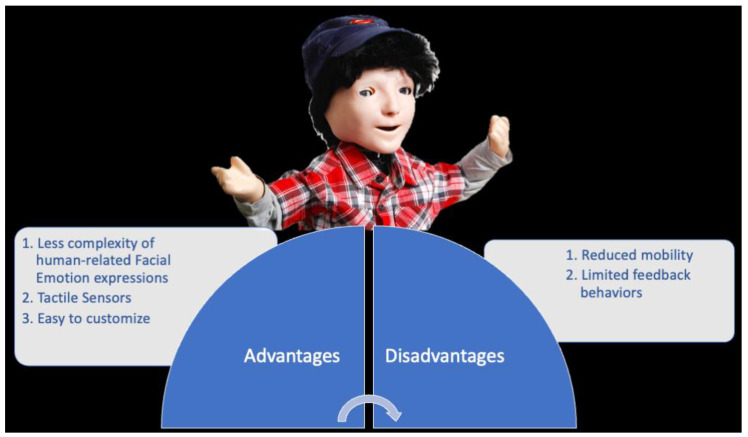
KASPAR.

**Figure 7 children-09-00953-f007:**
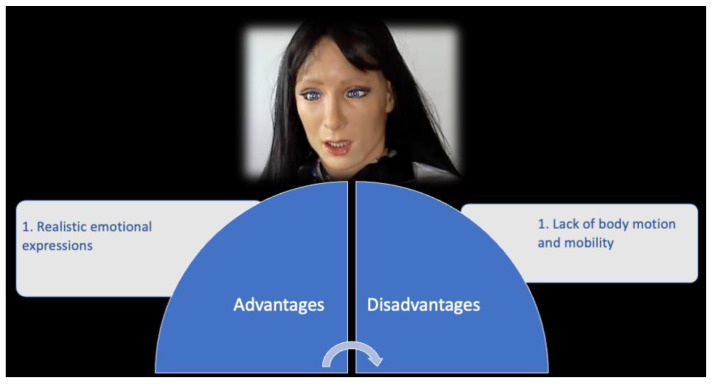
FACE robot.

**Figure 8 children-09-00953-f008:**
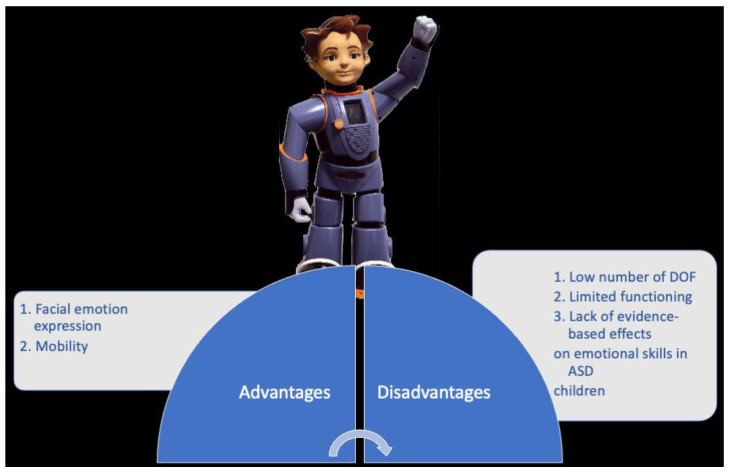
ZENO.

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
