# Peer review of "Social Humanoid Robots for Children with Autism Spectrum Disorders: A Review of Modalities, Indications, and Pitfalls"

_children, 2022, doi:10.3390/children9070953_

Round 1

Reviewer 1 Report

A review article, novel in the aspect of comparative analysis. The comparison of therapeutic tools in the form of humanoid robots with the division into advantages and disadvantages of particular robots in the aspect of supporting the development of children with ASD has been carried out correctly.

 The fragment of the text in which the authors cite the term "Valley of the Uncanny" requires clarification in the text what exactly it means and who is the creator of this concept. Conclusions properly constructed. 

The article is an excellent prelude to the research on the topic: the possibility of therapeutic effects in specific areas of individual robots on children with ASD - a comparative analysis.

Author Response

REPLY: we would like to thank this reviewer for this kind evaluation. The "Valley of the Uncanny" concept has been reformulated accordingly.

Reviewer 2 Report

The authors wrote an excellent review entitled "Social humanoid robots for children with autism spectrum disorders: A Review of Modalities, Indications, and Pitfall". The review is great and fully describes the problem, however, figures and figure legends have very pure quality and must be improved, especially:

1) Figures 2,3,4,6,7,8 have a low text resolution and ugly black background. Also, authors need to improve figure legends.

Author Response

we would like to thank this reviewer for this kind evaluation. All figures have been improved accordingly to the reviewer’s suggestion.